# Remote Spectral Light Sensing in the Home Environment: Further Development of the TWLITE Study Concept

**DOI:** 10.3390/s23084134

**Published:** 2023-04-20

**Authors:** Christina L. Reynolds, Aylmer Tan, Jonathan E. Elliott, Carolyn E. Tinsley, Rachel Wall, Jeffrey A. Kaye, Lisa C. Silbert, Miranda M. Lim

**Affiliations:** 1Department of Neurology, Oregon Health & Science University, Portland, OR 97239, USA; 2School of Medicine, Oregon Health & Science University, Portland, OR 97239, USA; 3VA Portland Health Care System, Research Service, Portland, OR 97239, USA; 4VA Portland Health Care System, Neurology, Portland, OR 97239, USA; 5Department of Behavioral Neuroscience, School of Medicine, Oregon Institute of Occupational Health Sciences, Oregon Health & Science University, Portland, OR 97239, USA; 6VA Portland Health Care System, Mental Illness Research Education and Clinical Center, National Center for Rehabilitative Auditory Research, Portland, OR 97239, USA

**Keywords:** tunable light, light sensor, sleep, smart living applications

## Abstract

Aging is a significant contributor to changes in sleep patterns, which has compounding consequences on cognitive health. A modifiable factor contributing to poor sleep is inadequate and/or mistimed light exposure. However, methods to reliably and continuously collect light levels long-term in the home, a necessity for informing clinical guidance, are not well established. We explored the feasibility and acceptability of remote deployment and the fidelity of long-term data collection for both light levels and sleep within participants’ homes. The original TWLITE study utilized a whole-home tunable lighting system, while the current project is an observational study of the light environment already existing in the home. This was a longitudinal, observational, prospective pilot study involving light sensors remotely deployed in the homes of healthy adults (*n* = 16, mean age: 71.7 years, standard deviation: 5.0 years) who were co-enrolled in the existing Collaborative Aging (in Place) Research Using Technology (CART) sub-study within the Oregon Center for Aging and Technology (ORCATECH). For 12 weeks, light levels were recorded via light sensors (ActiWatch Spectrum), nightly sleep metrics were recorded via mattress-based sensors, and daily activity was recorded via wrist-based actigraphy. Feasibility and acceptability outcomes indicated that participants found the equipment easy to use and unobtrusive. This proof-of-concept, feasibility/acceptability study provides evidence that light sensors can be remotely deployed to assess relationships between light exposure and sleep among older adults, paving the way for measurement of light levels in future studies examining lighting interventions to improve sleep.

## 1. Introduction

Normal aging is associated with changes in sleep patterns such as increasing sleep fragmentation, difficulty falling asleep, and potentially changes in sleep architecture [1]. These are often associated with, but not completely explained by other associated qualities of aging, such as increased medication usage and pain. Sleep quality is closely linked to cognitive function, with poor sleep quality affecting cognitive performance even in healthy aging adults [2]. This is of further concern because poor sleep quality is associated with a variety of health concerns, including an increased risk of falling, metabolic dysfunction, cardiovascular disease, stroke, mental health/depression, and long-term neurodegeneration [3,4].

One contributor to aging-related reductions in sleep quality could include inadequate and/or mistimed light exposure. Indeed, sleep disturbances impact cognitive health and appropriately timed light exposure is critical for optimal sleep and circadian rhythms—both of which become increasingly important in older age. Although acute lighting-based interventions are common, including in aging populations, there is a notable gap in the literature regarding the long-term continuous recording of light exposure. Moreover, research that combines these long-term continuous recordings of light exposure with additional objective physiologic outcomes would be especially impactful. Ultimately, all future long-term home-based lighting interventions should include a secondary means of unobtrusively continuously recording light levels, including the full-spectrum of wave lengths that may differentially impact sleep and circadian rhythms.

The present study demonstrates proof-of-concept data supporting the utility of light sensors for the long-term continuous recording of light exposure. These sensors were remotely deployed to participants already participating in an ongoing longitudinal project within the Oregon Center for Aging and Technology (ORCATECH) at Oregon Health and Science University (OHSU) called the Collaborative Aging Research using Technology (CART) study [5,6,7,8,9,10,11,12,13]. CART was developed as an interagency (National Institutes of Health and Department of Veterans Affairs) resource-related multicomponent project to develop and validate an infrastructure for digital health monitoring and intervention delivery that could be incorporated into the daily life of a diverse population of older adults [12]. Accordingly, this proof-of-concept study leveraged the suite of already installed in-home sensors, e.g., mattress-based sleep tracking, actigraphy, and passive infrared motion sensors. Primary outcomes include feasibility and acceptability for the continuous collection of light over a 12-week period.

## 2. Materials and Methods

### 2.1. Overview

The joint Institutional Review Board (IRB) between Oregon Health and Science University (OHSU; #17123) and the VA Portland Health Care System (VAPORHCS; #4089) approved this project. All participants provided written and verbal informed consent and were co-enrolled in the CART/ORCATECH study at OHSU [5,6,7,8,9,10,11,12,13].

Light sensors for this project were remotely deployed and installed within 10 homes across 16 participants (6 female, 10 male). Due to the prevalence of individuals with dementia enrolled in CART, and the cognitive requirement this project required, our primary a priori exclusion criterion was the presence of dementia. Therefore, all participants were adults without dementia between 60 and 80 years of age (mean age = 71.7 years; standard deviation = 5.0 years). As part of CART, data on long-term home activity (e.g., passive infrared motion sensor firing, sleep metrics, and wrist based actigraphy) from this in-home assessment platform were collected.

The present study is an extension of the previous Tunable White LIghT for Elders (TWLITE) study [14]. In TWLITE, participants homes were retrofitted with a tunable lighting system that was remotely controlled, such that all emitted light mirrored their natural environment according to time of day. An external light sensor broadly capturing spectral light information within participants home was absent from TWLITE. Accordingly, this pilot project employs such a light sensor and via leveraging the existing CART/ORCATECH infrastructure, associates light exposure with information about the movement and sleep patterns of individuals in the home to create a more accurate estimate of the light exposure experienced by study participants at home. The previously employed tunable lighting system was not concurrently implemented in the present study. All data were collected between June and September (zip code 972**). 

### 2.2. Light Sensors 

Light exposure within the home was monitored using the luxometer built into Actiwatch Spectrum Pro devices (Philips Respironics, Bend, OR, USA), which measure light (as photopic illuminance; Lux) in four spectral wavelengths: white light (380–750 nm), red light (600–700 nm), green light (500–600 nm), and blue light (400–500 nm), i.e., “RGB” spectrum. Devices automatically filtered each wavelength requiring no post-processing/filtering. The wristband was removed from each Actiwatch Spectrum Pro and off-wrist detection was disabled so that light data would be collected continuously while Actiwatches were in a static state. Four light sensors were installed in each home, with one sensor each attached to a wall in the kitchen, bedroom, bathroom, and living room (*n* = 40 sensors across 10 homes).

A 100% remote deployment method was utilized, and study participants were asked to install the light sensors themselves. Equipment and instructions were mailed to participants. Participants were instructed to affix each light sensor at eye-level to one wall across 4 different rooms of their home (e.g., kitchen, bedroom, living room, bathroom), in a central location at eye level, not adjacent to a window or light fixture. Sensors remained in place for 12 continuous weeks between June and September 2021.

### 2.3. Passive Infrared Motion Sensors 

In-home passive infrared (PIR) motion sensors (NYCE Sensors; Burnaby, BC, Canada) were used to estimate participant room location and general activity. PIR sensors were affixed to the wall in four common rooms within homes: bathroom, bedroom, kitchen, and living room, as previously described [5,12,13]. Each room was assigned a unique identifier, and thus, the firings of PIR sensors in rooms recorded the presence of a participant within a specific room. An example of PIR sensor firing events in four rooms is illustrated in Figure 1.

### 2.4. Time in Bed

Pressure-sensitive bed mats placed under the mattress (EmFit Corp., San Marcos, TX, USA) were used to record information about the patterns of study participants time in bed. These bed mats were previously validated against wrist-based actigraphy [11,15] as well as the PIR motion sensor network [14] to identify sleep; however, henceforth, “time in bed” will be retained as the primary bed mat-related outcome. Figure 2 shows the nightly time in bed for three representative study participants as measured using the EmFit bed mats. Over the course of this 12-week study, the presence of being in bed was detected on 92% of the nights. Nights with missing bed mat data generally aligned with 24 h periods of inactivity detected via the PIR sensor network, suggesting this was due to participants being out of the home and not from a device failure/interruption.

### 2.5. Actigraphy 

Wrist-based actigraphy was used to assess daily activity, determined via number of total steps (Withings Activite; Withings; Issy-les-Moulineaux, France). This actiwatch was validated in previous CART/ORCATECH studies [6] as well as the Ecologically Valid Ambient Longitudinal and Unbiased Assessment of Treatment Efficacy in Alzheimer’s Disease (EVALUATE-AD) study [16]. The Withings Activite watch was also previously validated against the Actigraph wGT3X-BT (Pensacola, FL, USA) and StepsCount PiezoRxD (Deep River, ON, Canada) [17]. Daily steps for three study participants are plotted in Figure 3. Participants reported robust usage, wearing the actiwatch on average 77% of days over this 12-week study period. 

### 2.6. Primary Outcomes and Data Analysis

The primary outcomes were feasibility and acceptability of the remote deployment of long-term light sensor installation. Feasibility outcomes included rates of recruitment, rates of withdrawal, reasons for participating, reasons for withdrawal, duration of participation, rates of equipment return, and rates of equipment failure. Participants were administered a standard acceptability questionnaire at the end of the 12-week study. This questionnaire included 7 items on a 4 point Likert scale, producing a total score of 28 (higher scores = greater acceptability).

Data from light sensors, PIR motion sensors, bed mats, and actigraphy were assessed as exploratory analyses. Data from light sensors were examined for fidelity, including missingness and proportion of unanticipated values (see Section 3). Light exposure within each of the 4 spectral wavelengths received was evaluated based on room location derived from PIR motion sensor data. Correlations between said light exposure and sleep/activity were evaluated; however, this pilot project did not have statistical power to draw conclusions on these potential relationships. Quality control/assurance measures for data collected included excluding light levels >4 standard deviations from the mean and confirming that missing bed mat/actigraphy data were not due to hardware malfunction. All data collected within this project were curated and publicly shared via dataverse (https://doi.org/10.7910/DVN/8GFDJJ).

## 3. Results

### 3.1. Feasibility of Light Sensors

A total of 20 homes were approached for participation in the study for a target enrollment of 10 homes, resulting in a recruitment rate of 50%. Of the 10 homes and 16 participants, none withdrew participation during the study. Reasons for participating included the desire to contribute to research, low burden, and prior positive experiences with CART/ORCATECH studies. Reasons for not participating included upcoming travel, surgery/medical procedures, or disinterest. All 10 homes returned equipment at the end of the 12-week study; however, one home delayed sending their equipment and post-surveys by one month due to unanticipated life events. This home only returned three of the four light sensors, resulting in overall rate of equipment return of 97.5%. All *n* = 40 light sensors collected data continuously for 12 weeks with no failures or interruptions in data collection.

### 3.2. Acceptability of Light Sensors

All participants were able to successfully install the sensors as instructed without issues. At the conclusion of the study, participants completed a survey about the acceptability of the light exposure sensors. All responses ranged from “Agree” to “Strongly Agree” (Table 1). Accordingly, overall acceptability was considered highly favorable with all participants reporting “Agree” or “Strongly Agree” across all questions.

### 3.3. Algorithm for Light Exposure in Monitored Rooms

Data were collected unimpeded in five-minute bins over the 12-week period from all 10 homes in four color bands (white, red, green, and blue). This produced 1,885,020 data points (no missing data were present). A small number of data points (0.42%) were found to have values higher than five standard deviations above the mean and were excluded from analysis.

The combination of light monitoring carried out using the Actiwatch Spectrum Pro light sensors and the PIR motion sensors made it possible to estimate the light exposure experienced by an individual in the rooms equipped with light sensors. Figure 4 illustrates the method of combining data from the PIR motion sensors and the Actiwatch light sensor to build an understanding of the light exposure experienced by a study participant in his or her home environment. The top panel of the figure displays the motion detections by four PIR sensors over a two-day period, giving an estimate of the participant’s movements within the home.

The subsequent four panels of Figure 4 show the white light lux measured by the Actiwatch light sensors in the same four rooms over the same two-day period. Light values were recorded every five minutes. For each motion detection, a light sensor value was found in the same room as the motion detector firing event with five minutes of the motion detection. These values are marked with red circles.

The Actiwatch Spectrum Pro light sensor (Philips Respironics; Bend, OR, USA) was capable of measuring irradiance in three color bands: red (600–700 nm), green (500–600 nm), and blue (400–500 nm). This makes it possible to determine not just the total amount of home light exposure an individual experienced throughout the day. The color of light can also be characterized. Figure 5 illustrates the total daily irradiance in three colors for the same study participant over the entire twelve weeks of the study. These values were obtained using the same approach combining motion sensor firings and light sensor values previously described for white light [14]. An example of combining these two data sets to estimate the in-home light exposure for an individual living alone is shown in Figure 5. A plot of the ratio of total red light exposure to blue light exposure is displayed in Figure 6, providing an example of the possible calculations that can be carried out with this data which may be of value in sleep and circadian rhythm research. Of note, global assessments of potential correlations between each spectral wavelength and time in bed (derived via bed mats, actigraphy, and PIR) were not significantly correlated. This may be attributed to the sample size or light exposure received outside of the home.

## 4. Discussion

In this pilot study, we demonstrated feasibility and acceptability for the remote installation of sensors to collect light exposure from participants’ native home environments. The results indicate high feasibility and acceptability for this approach. The results also demonstrate high data fidelity with the present equipment. Data collected from this pilot study also show promise for exploring both the relationship between light exposure and sleep duration as well as daily activity. This future tool will aid in the development of lighting interventions for the treatment of sleep and circadian rhythms disorders, especially in the population of older adults for whom these are significant issues.

Light exposure was shown to impact sleep and daily rhythms, but few studies examined light exposure continuously in the native home environment [18,19,20,21]. Advantages to our light sensor protocol included wireless configuration with long battery life (e.g., 12 weeks between charges), but they also allowed us to measure red, green, blue (RGB) spectral bands, in addition to white light. Measures of activity and time in bed from the sensors included in this study are presented in Table 2. The availability of this data time-aligned with the light exposure calculated in Section 3 is highly useful for future studies of sleep patterns and circadian rhythms.

There are few effective current pharmacological and non-pharmacological therapies for sleep disturbances. Sleep hygiene practices, including limiting caffeine and alcohol intake, avoiding evening light exposure from computers or the television, exercising regularly, and keeping regular bedtimes and wake times with adequate light exposure upon waking, are a start, but limited in efficacy and adherence [22,23]. Aging can present unique challenges in improving sleep and delivery of light as a potential therapy. Aging is associated with inactive suprachiasmatic nucleus cells, which control circadian rhythms, that can be reversed with exposure to bright light, as shown in a rat model of aging [24]. Thus, more regular light exposure could help better entrain dysfunctional circadian rhythms, and possibly improve physical activity during the day [25]. Prior studies differed in the delivery, timing, dosage, and duration of light therapy, with variable contributions from specific wavelengths, as well as variable outcomes including sleep, circadian realignment, and cognition, as well as mood [26]. Future studies examining lighting interventions to improve sleep would benefit from in-home continuous long-term recording of light levels across the RGB spectra.

A number of limitations were inherent in this study. As a pilot study, the sample size was small with only sixteen participants living in 10 homes. Nevertheless, this sample size was adequate to draw conclusions about feasibility, acceptability, and data fidelity of the devices. The study also did not consider factors such as geographical latitude, weather, seasonal effects on environmental light, or window location relative to the sensor placement. However, participants were instructed not to place light sensors adjacent a window or immediately next to a light fixture; ideally in a central location on a wall. With this in mind, even if light sensors were placed next to a window or light fixture, this still did not control for the use of blinds and specific light fixtures. Future studies may opt to employ a larger number of light sensors to produce a better overall within room average. Similarly, this study could not control for light levels received outside of the home. To adequately address this in future, projects will involve participants wearing a light sensor 24/7.

Other limitations were due to the device chosen; while simple and unobtrusive in design and wireless, the Actiwatch Spectrum Pro devices have a limited battery life (12 weeks), and data cannot be assessed in real-time. For a longer-term continuous study, it would be necessary to integrate WiFi or Bluetooth connectivity to light exposure, sleep duration, and activity measurement sensors to facilitate collection and assessment of data in real-time, on an ongoing basis. In order to scale this study to larger populations, the Actiwatch Spectrum Pro may need to be replaced with a more economical option in order to include more homes and monitor more locations within each home. The use of motions sensors to determine the location of a participant within the home impose another limitation. It is not possible to differentiate between multiple individuals using the motion sensors, and so, this method is best suited to single person homes.

## 5. Conclusions

This pilot study demonstrated the feasibility and acceptability of long-term, continuous light exposure monitoring in the home environment of older adults, a population especially impacted by changes in sleep patterns and sleep disorders. The approach of using 100% remote deployment is feasible and acceptable. The foundational work completed during this study can be expanded in future research to support the validation of light-based interventions for sleep and circadian rhythm disorders.

## Figures and Tables

**Figure 1 sensors-23-04134-f001:**
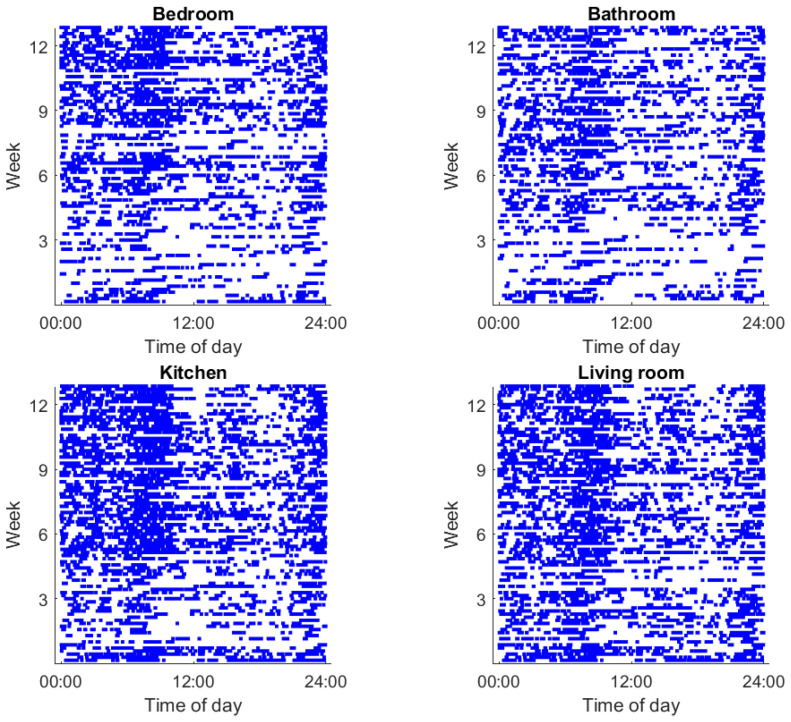
PIR motion sensor firings indicating motion within a given rooms, with each point recorded as a specific time and date. These observations allow for the unobtrusive determination of the times when each of these rooms within the home are occupied. Blue periods correspond to motion being detected, whereas open periods correspond to periods without motion.

**Figure 2 sensors-23-04134-f002:**
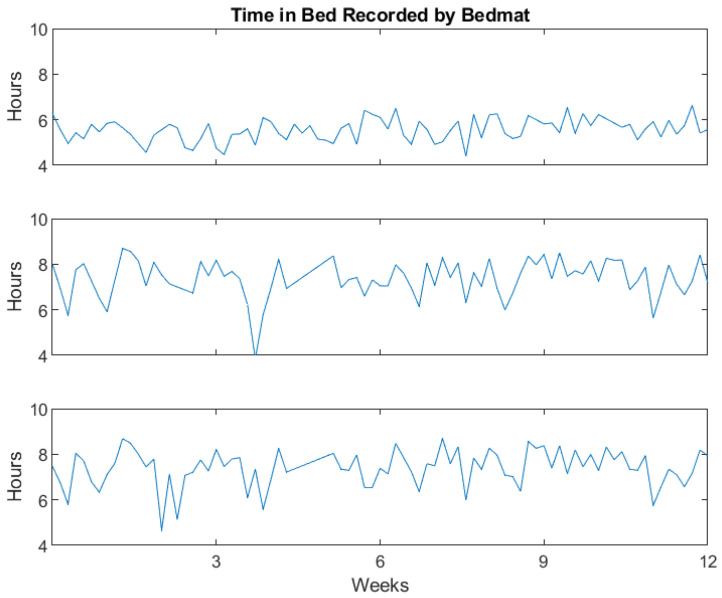
Nightly total time in bed for three representative participants using the Emfit bed mat.

**Figure 3 sensors-23-04134-f003:**
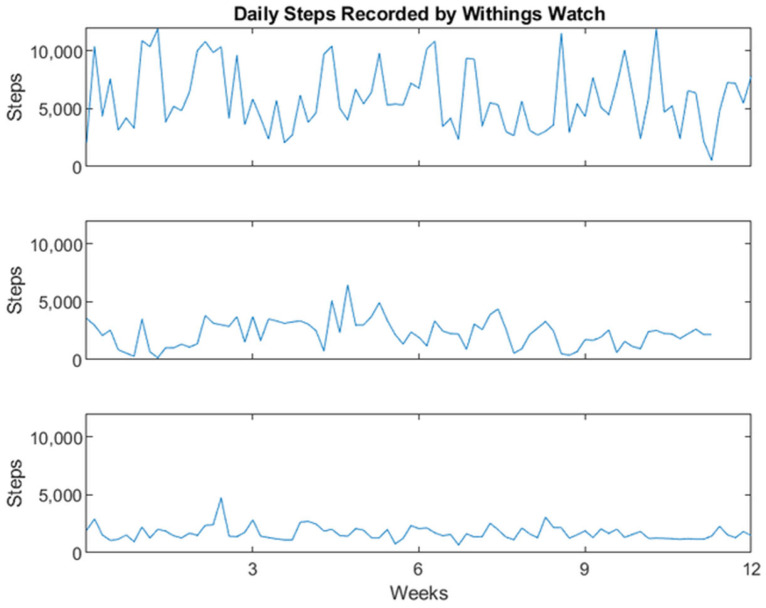
Daily total step counts for three representative study participants as measured using the Withings Activite watch.

**Figure 4 sensors-23-04134-f004:**
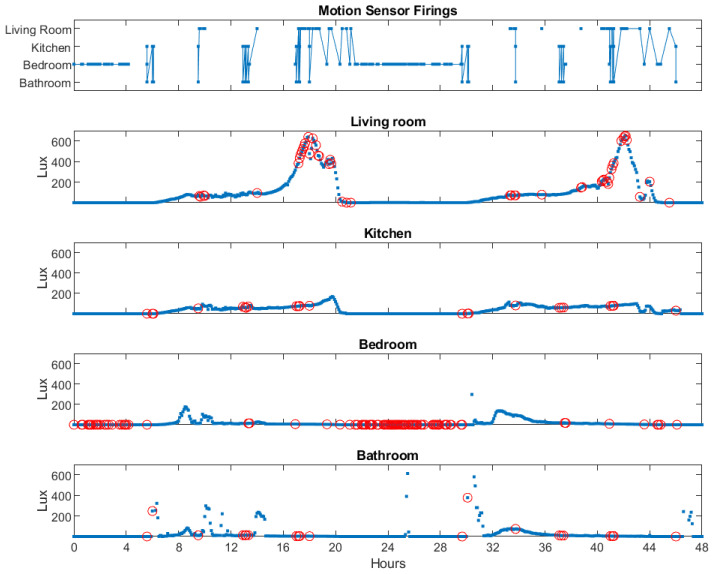
Top panel: motion sensor firings in a single person home over a two-day (48 h) period, showing motion detection events in four rooms of the home. Gaps in data represent sedentary behavior or the individual not being present in one of these four rooms. The second through fifth panels show the white light measurements taken every five minutes by an Actiwatch Spectrum Pro sensor in each of four rooms, with measurements corresponding to motion sensor firings marked with red circles.

**Figure 5 sensors-23-04134-f005:**
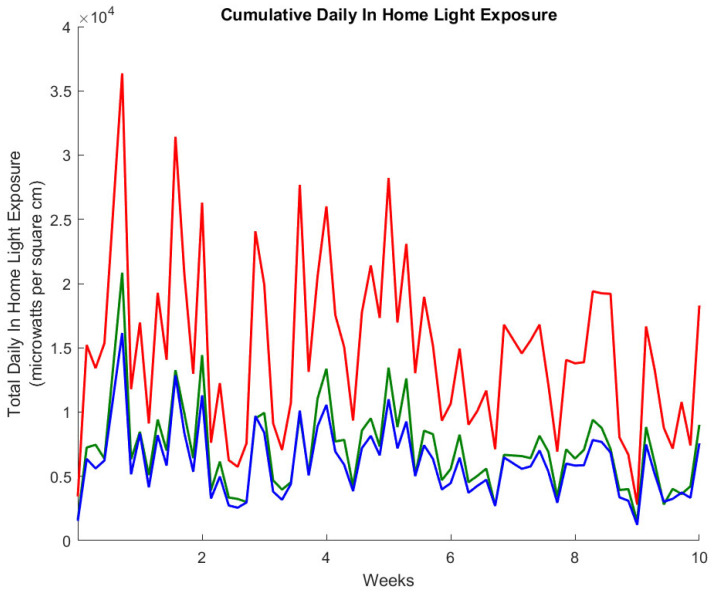
Cumulative daily in-home light received by a participant, factoring in specific room locations within the home: total daily in-home exposure (microwatts per centimeter squared) summed from the four rooms of the home of the study participant, using only light measurements from rooms when motion was detected within a five-minute period in that room. Color bands are defined as red (600–700 nm), green (500–600 nm), and blue (400–500 nm).

**Figure 6 sensors-23-04134-f006:**
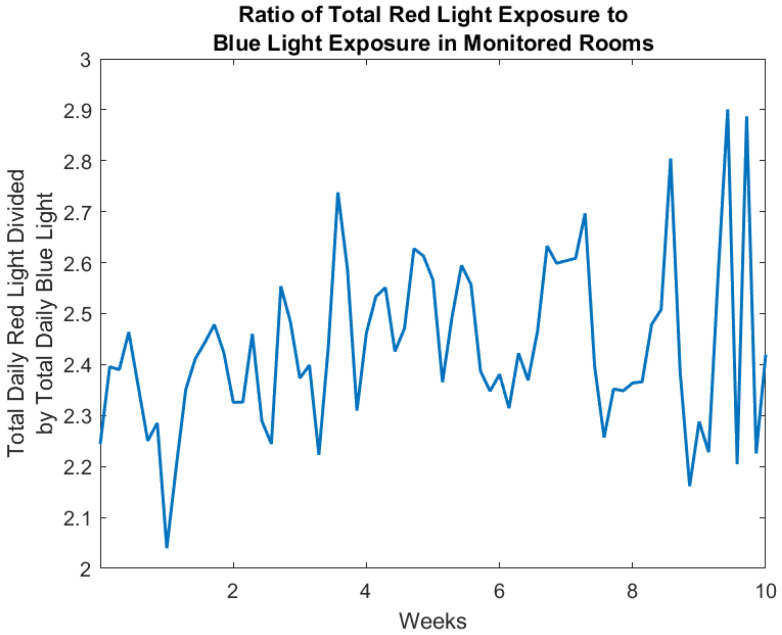
Ratio of daily in-home red light exposure in microwatts per square centimeter (600–700 nm) to daily in-home blue light exposure in microwatts per square centimeter (400–500 nm) summed from the four rooms of the home of the study participant, using only light values when motion was detected in the room.

**Table 1 sensors-23-04134-t001:** User satisfaction and acceptability survey results.

Survey Question	Response
1. The instructions were easy to follow	3.7 ± 0.5
2. I received answers to all questions I had	3.4 ± 0.8
3. I feel like I completed the sensor protocol properly	3.4 ± 0.5
4. I feel like my involvement in testing these sensors was valued	3.0 ± 0.8
5. The equipment was easy to use	3.3 ± 0.5
6. The equipment was unobtrusive and seemed to function as expected	3.3 ± 0.5
7. I would use this equipment again for a longer duration in the future	3.1 ± 0.7
**Average Total Score (out of 28):**	**23.3 ± 3.5**

Data are presented as mean ± standard deviation. Response options were: 0: Strongly Disagree, 1: Disagree, 2: Neither Agree nor Disagree, 3: Agree, 4: Strongly Agree.

**Table 2 sensors-23-04134-t002:** Sensor specifications for activity and sleep/time in bed.

Sensor	Specifications
**Withings Activite**	
Battery life	Up to 18 months
Activity metrics	Total daily steps, time and duration of each activity session, number of steps in each active session, time of first and last activity each day
**Emfit QS Bed Mat**	
Sleep metrics	Daily duration of time in bed, daily duration of time asleep and awake in bed, wake after sleep onset (WASO), number of awakening and bed exits
Sleep event times	Time and date of start and end of each period in bed and asleep, time and date of each awakening and bed exit
Physiological data	Heart rate variability and respiratory rate

## Data Availability

A de-identified, anonymized dataset will be created and shared, and released once institutional approvals are in place. Where practicable, sharing will take place under a written agreement prohibiting the recipient from identifying or re-identifying (or taking steps to identify or re-identify) any individual whose data are included in the dataset. However, it is permissible for final datasets in machine-readable format to be submitted to and accessed from PubMed Central (and similar sites) provided that care is taken to ensure that the individuals cannot be re-identified using other publicly available information. De-identified data sets will be maintained locally on VA computer drives that are regularly backed up and password protected until central data repositories become available for long-term storage and access. Until that time, the data sets will be available by request to any interested investigators together with a data dictionary to enable interpretation of data set.

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
