# Peer review of "Remote Spectral Light Sensing in the Home Environment: Further Development of the TWLITE Study Concept"

_sensors, 2023, doi:10.3390/s23084134_

Round 1
Reviewer 1 Report
The title differs from the title in the system.
The manuscript title is A protocol for remote spectral light sensing and sleep assessment in the home environment: TWLITE+, a follow-on protocol to TWLITE.
In the system, the title is Tunable White Light for Elders (TWLITE+): A pilot study of deployment, remote data collection, and analysis of light exposure, sleep, and activity in the home environment.
The manuscript is reviewed using the title of the manuscript.
With this title in mind, a description of a protocol is expected. This, however, is not the case.
The introduction of the manuscript is describing how light might influence the sleep of people with Alzheimer’s disease. One keyword is also Alzheimer’s disease. This is not appropriate at all, since no people with Alzheimer’s disease or the dementia syndrome were part of the study. Therefore, this keyword must be replaced and the introduction should elaborate on the influence of light on older people. The authors could, however, discuss the possible influence of light in the discussion section when reflecting on their findings. The authors should focus on the introduction of the manuscript on the rationale of the study namely: Primary outcomes include feasibility, acceptability, and data fidelity for the continuous 62 collection of light over a 12-week period.
In the material and method section, the inclusion and exclusion criteria of the participants are missing. Only non-demented is given as an exclusion criterion. This is a non-professional term to refer to people with dementia. The rationale for this exclusion criteria is lacking.
In this study, light exposure at home is measured. The light exposure of the participant at home will also be influenced by the overall light exposure during the day. How was this taken into account? Therefore it is also relevant to provide the period in the year and the geographical location of the study. No information is given about the designed lighting program used in this study.
Also, no description is given about the instruction that the participants received. This should be added to the material and method section or should be at least added in the appendix.
No description is given about how Light exposure levels within each of the 4 spectral wavelength light bands were correlated with sleep (estimated from bed mats) and activity/step count (derived from wrist-based actigraphy) as well as room location (determined by PIR). How were the measurements validated?
All available data regarding the measurements of the 16 participants and 10 homes should be provided for transparency and reproducibility of the study. So please provide all data regarding the sleep time/steps for all participants in a table. No description of the parameters for acceptability is given. Which method or test is used? Please provide a link to the validated questionnaires.
Figure 5 illustrates the total daily exposure based on the sensors at home. Please describe in the method section how this is calculated while taking into account the light exposure outdoors. No description of the algorithms used is provided
The manuscript is lacking a description of a protocol. With the current data, the study is not reproducible for others.
The description of sensors’ specifications should be described as part of the method section.
The conclusion is given that Pilot data collected during the twelve-week study suggests that daily light exposure is correlated with both sleep duration and daily activity. Daily activity is restricted to step counts.
Author Response
Reviewer #1
Thank you for the careful and thoughtful review of our manuscript. Detailed responses follow each point, however, in summary, we have 1) revised the paper (including the title) to more appropriately reflect the study design lessening the emphasis on this being a strict “protocol” publication, 2) focused the introduction, removing reference to Alzheimer’s Disease, 3) clarified the assessment of light exposure, including added detail on the specific light sensors, and 4) provided the complete raw data set for full transparency, and created an appendix describing participant instructions.
- The title differs from the title in the system.
RESPONSE: This has been corrected in the system. The title is: “Remote spectral light sensing in the home environment: Further development of the TWLITE study concept.”
- With this title in mind, a description of a protocol is expected. This, however, is not the case.
RESPONSE: The reviewer makes an excellent point, see above for our revised title. Furthermore, we have revised the manuscript to lessen the focus on this being a traditional “protocol” manuscript, as it is not.
- The introduction of the manuscript is describing how light might influence the sleep of people with Alzheimer’s disease. One keyword is also Alzheimer’s disease. This is not appropriate at all, since no people with Alzheimer’s disease or the dementia syndrome were part of the study. Therefore, this keyword must be replaced and the introduction should elaborate on the influence of light on older people. The authors could, however, discuss the possible influence of light in the discussion section when reflecting on their findings. The authors should focus on the introduction of the manuscript on the rationale of the study namely: Primary outcomes include feasibility, acceptability, and data fidelity for the continuous collection of light over a 12-week period.
RESPONSE: We have removed Alzheimer’s Disease as a keyword, and reference to Alzheimer’s has been removed from the introduction which now remains focused on older individuals in the context of the stated study rationale. As suggested, we have included a short comment regarding the potential association between light, sleep and neurodegeneration (including Alzheimer’s) in the discussion.
- In the material and method section, the inclusion and exclusion criteria of the participants are missing. Only non-demented is given as an exclusion criterion. This is a nonprofessional term to refer to people with dementia. The rationale for this exclusion criteria is lacking.
RESPONSE: We have expanded our inclusion/exclusion criteria description in the methods section, and clarified that participants were required to not be presently diagnosed with dementia. The rationale for this exclusion was due to the large number of participants in this broader ORCATECH cohort having dementia and the requirement for this study that participants install these sensors themselves; potentially a cognitively challenging task for someone with dementia.
- In this study, light exposure at home is measured. The light exposure of the participant at home will also be influenced by the overall light exposure during the day. How was this taken into account? Therefore it is also relevant to provide the period in the year and the geographical location of the study. No information is given about the designed lighting program used in this study.
RESPONSE: The present study did not measure light received outside of participants homes. While a crude estimate of total time spent out of the home was able to be determined via PIR motion sensors, this still does not measure light received outside of the home. Which as the reviewer notes will remain a limitation short of participants wearing a light sensor 24/7. We have added additional discussion addressing this limitation.
We had omitted including dates and specific geographical location due to potential PHI issues, however, we agree this is important and can specify the study range was from June-September in the 972** zip code.
To clarify, the homes involved in this study did not have an active lighting intervention (i.e., were not simultaneously engaging in TWLITE; reference #14). This point has been clarified in the methods section and throughout the manuscript.
- Also, no description is given about the instruction that the participants received. This should be added to the material and method section or should be at least added in the appendix.
RESPONSE: Thank you, we have included a more detailed description of participant instructions in the methods section.
- No description is given about how Light exposure levels within each of the 4 spectral wavelength light bands were correlated with sleep (estimated from bed mats) and activity/step count (derived from wrist-based actigraphy) as well as room location (determined by PIR). How were the measurements validated?
RESPONSE: Thank you, this is an excellent observation. We carefully reviewed these data, correlating each spectral wavelength with sleep (via bedmats, actigraphy, and PIR), and found no relationships. This could be attributed to the generally small sample size or potentially even confounded from light levels received outside of the home (as the reviewer pointed out in an earlier comment). Further description of this has been included in the manuscript.
- All available data regarding the measurements of the 16 participants and 10 homes should be provided for transparency and reproducibility of the study. So please provide all data regarding the sleep time/steps for all participants in a table. No description of the parameters for acceptability is given. Which method or test is used? Please provide a link to the validated questionnaires.
RESPONSE: Assessment of acceptability was completed through a custom questionnaire (Table 1), providing a semi-quantitative description. Standard Likert scale questions were designed according to the theoretical framework for acceptability principles (Sekhon, Cartwright, and Francis. BMC Health Services Research, 2017) addressing affective attitude, burden, perceived effectiveness, ethicality, intervention coherence, opportunity costs, and self-efficacy.
We agree that data sharing is crucial and have compiled all study related data for each subject (i.e., light, actigraphy, bedmat, PIR, etc.), and made this file accessible to the public via Dataverse (an ORCID connected open-access data sharing platform; https://dataverse.org/researchers). The URL to this data set has been included in the manuscript and will be fully accessible upon publication.
- Figure 5 illustrates the total daily exposure based on the sensors at home. Please describe in the method section how this is calculated while taking into account the light exposure outdoors. No description of the algorithms used is provided.
RESPONSE: Thank you for highlighting this clarification. As described in the response to point #5, light exposure received outside of the home was not quantified. We have clarified this throughout the manuscript, and revised figure titles to be more specific (e.g., Figure 5; Cumulative Daily In-home Light Received).
- The manuscript is lacking a description of a protocol. With the current data, the study is not reproducible for others.
RESPONSE: We refer the reviewer to our response to point #2.
- The description of sensors’ specifications should be described as part of the method section.
RESPONSE: We have added that the Philips Spectrum Pro records light as photopic illuminance (lux) across the stated wave lengths (range 380-750 nm).
- The conclusion is given that Pilot data collected during the twelve-week study suggests that daily light exposure is correlated with both sleep duration and daily activity. Daily activity is restricted to step counts.
RESPONSE: This has been removed.
Reviewer 2 Report
The authors present a study protocol for remote light sensing and sleep assessment in the home environment. The proposed protocol TWLITE+ follows-on the previous protocol TWLITE and extends its capabilities.
The authors introduce the pilot project which is the base for the proposed study, and used sensing methods for light exposure, motion, sleep and daily activity. The outcomes of the study are feasibility and acceptability of the installation. The obtained data was exploratory analysed. The results were evaluated on the basis of a questionnaire survey conducted in 10 homes (16 participants) at the end of 12-week study.
The paper is well prepared and organized; I recommend accepting the paper.
Minor Comments:
1. lines 27 and 80: Please, report the standard deviation with the same number of decimal places as the mean value.
Author Response
Reviewer #2
The authors present a study protocol for remote light sensing and sleep assessment in
the home environment. The proposed protocol TWLITE+ follows-on the previous protocol TWLITE and extends its capabilities.
The authors introduce the pilot project which is the base for the proposed study, and used
sensing methods for light exposure, motion, sleep and daily activity. The outcomes of the study are feasibility and acceptability of the installation. The obtained data was exploratory analysed. The results were evaluated on the basis of a questionnaire survey conducted in 10 homes (16 participants) at the end of 12-week study.
The paper is well prepared and organized; I recommend accepting the paper.
Thank you.
Minor Comments:
- lines 27 and 80: Please, report the standard deviation with the same number of decimal places as the mean value.
RESPONSE: The SD values on lines 27 and 80 have been updated to match the decimal places as the previously reported mean values.
Reviewer 3 Report
Comments to the authors:
Manuscript title: A protocol for remote spectral light sensing and sleep assessment in the home environment: TWLITE+, a follow-on protocol to TWLITE
Overall summary: Here, the authors Reynolds et al evaluate TWLITE+ as proof-of-concept pilot study to measure actigraphy, static light measures, and location with movement detectors within homes. They describe the overall study design and set-up, with 1) feasibility and 2) acceptability of remote deployment as the primary outcomes. The overall study concept is novel and an important contribution to the literature; however, the authors make some unsupported claims that should be amended.
Additional reviewer comments are listed by manuscript section, below:
General comments:
Throughout the text, the authors claim to have measured “sleep”; however, they used mattress-based activity sensors, without any collection of sleep diary data. Therefore, there is no way to accurately determine sleep onset and offset to calculate sleep duration. Rather, the authors measured time in bed. Please update the entire manuscript accordingly to reflect that time in bed was the metric that was measured.
Abstract: Please include sample size information and that light was measured with ActiWatch Spectrum devices.
Introduction:
- Introduction describes aging, sleep, and light, but the paper is about proof-of-concept for remotely monitoring people’s activity in their home environments; please provide greater background on this
- Line 62: As stated in later comments, “data fidelity” is mentioned but results aren’t provided to support this
Materials and methods:
- Line 79: please replace the term “non-demented” with people-first language – e.g., “without dementia / symptoms of dementia”
- Please include a section describing QA/QC methods for the analyzed data
Results:
- In lines 180-181, please move to methods and describe in greater detail; how were data filtered? What were the “expected” ranges? Also, define the term “RGB”
- What were the results when missingness and data quality were checked? What was the overall proportion of missing data / invalid data? Were there any factors that were identified that may have impacted this? This information would be useful to know for future studies
Discussion:
- Because sampling occurred from June-September, the change in cumulative light measures and light spectra is likely due to seasonal changes in light exposure from the transition of the summer solstice to the fall equinox.
- Lines 247-249: nothing is presented to support these comments, and thus they should be removed. There was no measurement of sleep duration or sleep time, no details / appropriate statistical analysis with adjustment for confounding factors (such as season), and no results presented.
- Line 281: Please name the other device initially tested that was not used
- Position of the light sensors in the house, such as whether they faced a window or not, would be expected to have large impacts on the overall measures; while this is briefly mentioned in the limitations, please expand on this and how future studies could address this issue.
- In the Figure 4 panels 2-5, there are episodes where the lux levels appear to change dynamically (perhaps from a room light being turned on or sunlight angle through a window), but the change is not always matched with movement in the room; for example, in the bedroom plot, there are two bumps near 8AM and 10AM where lux increases, but no detection of movement. Can you speculate what may have happened here?
Conclusion:
- Line 296: state “high data fidelity” but don’t provide the data to support this claim
- Remove sentence from lines 297-298
Figures and Tables:
- Figure 1: in the figure caption, please include information about what the blue/white colors are representing (does white represent activity that triggered the PIR?). Also, please add y-axis labels for each plot, with the same axis marks to easily compare across
- Additionally, for Figure 1, data from weeks 3-6 look sparser than data from week 6 onwards – can you speculate as to why? Did you collect information about whether the participant did any travelling / if they occupied the house for the entire study or if they took a trip somewhere?
- Figure 2: As mentioned in the general comments, please change the “Hours of Sleep” label and “total sleep time” caption to “time in bed”
Data Availability Statement: Data should be made accessible prior to publication. One potential data repository is the National Sleep Research Resource, supported by NHLBI: https://sleepdata.org/
Author Response
Reviewer #3
Here, the authors Reynolds et al evaluate TWLITE+ as proof-of concept pilot study to measure actigraphy, static light measures, and location with movement detectors within homes. They describe the overall study design and set-up, with 1) feasibility and 2) acceptability of remote deployment as the primary outcomes. The overall study concept is novel and an important contribution to the literature; however, the authors make some unsupported claims that should be amended.
Throughout the text, the authors claim to have measured “sleep”; however, they used mattress-based activity sensors, without any collection of sleep diary data. Therefore, there is no way to accurately determine sleep onset and offset to calculate sleep duration. Rather, the authors measured time in bed. Please update the entire manuscript accordingly to reflect that time in bed was the metric that was measured.
RESPONSE: We have updated the manuscript throughout to reflect “time in bed” rather than “sleep”.
Abstract: Please include sample size information and that light was measured with ActiWatch Spectrum devices.
RESPONSE: Done.
Introduction describes aging, sleep, and light, but the paper is about proof-of concept for remotely monitoring people’s activity in their home environments; please provide greater background on this
RESPONSE: Done.
Introduction Line 62: As stated in later comments, “data fidelity” is mentioned but results aren’t provided to support this
RESPONSE: The term “data fidelity” was removed from this line.
Methods: Line 79: please replace the term “non-demented” with people-first language – e.g., “without dementia / symptoms of dementia”
RESPONSE: The phrase “non-demented” has been replaced with “adults without dementia”.
Methods: - Please include a section describing QA/QC methods for the analyzed data
RESPONSE: Additional detail has been added to section 2.6 in the Methods section.
Results: In lines 180-181, please move to methods and describe in greater detail; how were data filtered? What were the “expected” ranges? Also, define the term “RGB”
RESPONSE: We have expanded on this section within the Methods section and defined RGB.
Results: What were the results when missingness and data quality were checked? What was the overall proportion of missing data / invalid data? Were there any factors that were identified that may have impacted this? This information would be useful to know for future studies
RESPONSE: We have described data missingness and data quality checking in greater detail in the manuscript, within the appropriate sub-section for each outcome. In short, there was very little missing data, e.g., time in bed was recorded on 92% of the nights with the remaining 8% of nights aligning with time spent outside of the home. There was no missing data from the PIR network or light sensors. There were however a small percentage (0.42%) of light sensors data that was >4 standard deviations away from the mean and excluded).
Discussion: Because sampling occurred from June-September, the change in cumulative light measures and light spectra is likely due to seasonal changes in light exposure from the transition of the summer solstice to the fall equinox.
RESPONSE: Thank you, this is a good point to highlight, which another reviewer also alluded to. This has been clarified.
Discussion: Lines 247-249: nothing is presented to support these comments, and thus they should be removed. There was no measurement of sleep duration or sleep time, no details / appropriate statistical analysis with adjustment for confounding factors (such as season), and no results presented
RESPONSE: This sentence has been deleted.
Discussion: Line 281: Please name the other device initially tested that was not used
RESPONSE: This was an error on our part, and this comment has been removed.
Discussion: Position of the light sensors in the house, such as whether they faced a window or not, would be expected to have large impacts on the overall measures; while this is briefly mentioned in the limitations, please expand on this and how future studies could address this issue.
RESPONSE: This has been expanded on within the limitations section and elsewhere.
In the Figure 4 panels 2-5, there are episodes where the lux levels appear to change dynamically (perhaps from a room light being turned on or sunlight angle through a window), but the change is not always matched with movement in the room; for example, in the bedroom plot, there are two bumps near 8AM and 10AM where lux increases, but no detection of movement. Can you speculate what may have happened here?
RESPONSE: This is an excellent observation. Possible speculation for these episodes would include a participant sitting stationary in a chair and turning on/off a lamp or overhead light, or possibly significant changes in environmental light due to changes in cloud cover. An important detail regarding the PIR motion sensors is that their signal should go silent without movement, in other words, they do not detect continual presence or a participant being sedentary.
Conclusion: Line 296: state “high data fidelity” but don’t provide the data to support this claim
RESPONSE: This phrase has been removed.
Conclusion: Remove sentence from lines 297-298
RESPONSE: Done.
Figure 1: in the figure caption, please include information about what the blue/white colors are representing (does white represent activity that triggered the PIR?). Also, please add y-axis labels for each plot, with the same axis marks to easily compare across
RESPONSE: Done.
Additionally, for Figure 1, data from weeks 3-6 look sparser than data from week 6 onwards – can you speculate as to why? Did you collect information about whether the participant did any travelling / if they occupied the house for the entire study or if they took a trip somewhere?
RESPONSE: Self-reported extended travel information or hospitalization (>1 day) records are available for subjects, however this participant did not report either of these. This time period corresponded to the summer months (June-August); it is possible this participant spent more time outdoors during this time. Alternatively, this participant may have also been more sedentary.
Figure 2: As mentioned in the general comments, please change the “Hours of Sleep” label and “total sleep time” caption to “time in bed”
RESPONSE: Done.
Data Availability Statement: Data should be made accessible prior to publication. One potential data repository is the National Sleep Research Resource, supported by NHLBI: https://sleepdata.org/
RESPONSE: Thank you, we agree with the reviewer. Per the request of another reviewer, we have chosen to make this full data set freely accessible via dataverse (URL/link to the data set is within the manuscript). The entire manuscripts dataset has been uploaded and will be fully accessible upon publication.
Round 2
Reviewer 1 Report
Thank you for the time to follow-up on the recommendations made.
I am satisfied with the adjustments made.
I only could not check the URL for data-sharing (All data collected within this project has been curated and 170 publicly shared via dataverse (https://doi.org/10.7910/DVN/8GFDJJ). 171 3.)
Author Response
Reviewer 1. Noted that our data sharing URL was inactive. The fully public URL will activate when this manuscript is accepted, per standard guidelines. Here is a private URL for accessing these data prior to publication that can be shared with this reviewer. Please let me know if barriers to access remain.
https://dataverse.harvard.edu/privateurl.xhtml?token=f2e3bb96-bd0a-468a-9a0e-5d57a8c6f6ea
Reviewer 3 Report
The authors have done a nice job editing and addressing the reviewer comments; I have no further feedback.
Author Response
no comments